# Photocurable 3D-Printable Systems with Controlled Porosity towards CO_2_ Air Filtering Applications

**DOI:** 10.3390/polym14235265

**Published:** 2022-12-02

**Authors:** Annalisa Chiappone, Alessandro Pedico, Stefania Porcu, Candido Fabrizio Pirri, Andrea Lamberti, Ignazio Roppolo

**Affiliations:** 1Dipartimento di Scienze Chimiche e Geologiche, Università di Cagliari, S.S. 554 bivio Sestu, 09042 Monserrato, Italy; 2Department of Applied Science and Technology, Politecnico di Torino, C.so Duca Degli Abruzzi 24, 10129 Turin, Italy; 3Center for Sustainable Future Technology Polito, Italian Institute of Technology, Via Livorno 60, 10144 Turin, Italy; 4Department of Physics, Università di Cagliari, S.p. no. 8 Km 0700, 09042 Monserrato, Italy

**Keywords:** 3D printing, photocurable emulsion, CO_2_ capture, air-filtering

## Abstract

Porous organic polymers are versatile platforms, easily adaptable to a wide range of applications, from air filtering to energy devices. Their fabrication via vat photopolymerization enables them to control the geometry on a multiscale level, obtaining hierarchical porosity with enhanced surface-to-volume ratio. In this work, a photocurable ink based on 1,6 Hexanediol diacrylate and containing a high internal phase emulsion (HIPE) is presented, employing PLURONIC F-127 as a surfactant to generate stable micelles. Different parameters were studied to assess the effects on the morphology of the pores, the printability and the mechanical properties. The tests performed demonstrates that only water-in-oil emulsions were suitable for 3D printing. Afterwards, 3D complex porous objects were printed with a Digital Light Processing (DLP) system. Structures with large, interconnected, homogeneous porosity were fabricated with high printing precision (300 µm) and shape fidelity, due to the addition of a Radical Scavenger and a UV Absorber that improved the 3D printing process. The formulations were then used to build scaffolds with complex architecture to test its application as a filter for CO_2_ absorption and trapping from environmental air. This was obtained by surface decoration with NaOH nanoparticles. Depending on the surface coverage, tested specimens demonstrated long-lasting absorption efficiency.

## 1. Introduction

The rising temperature of our planet and its consequences over the years are currently a prominent topic in scientific debate. According to some studies carried out by the Intergovernmental Panel on Climate Change, the main cause of global warming can be attributed to the release of carbon dioxide by human activities, thus increasing the greenhouse effect [1]. Carbon capture and sequestration (CCS) is fundamental for efficient carbon emissions management because it is able to selectively capture CO_2_ and store it permanently. Among the capture technologies available, porous materials offer many advantages, since they are an economical and practical solution due to their simplicity, design compactness, easy handling, material stability, low energy consumption and eco-friendliness, given the absence of solvents and other toxic chemicals during use [2]. Pore size and shape can be finely tuned to act as selective sieves to fine-tune capture chemistry in adsorbent materials, characterized by high uptake efficiency and high adsorption capacity as a result of their extended internal surfaces due to a significant surface-area-to-volume ratio [3]. In the last few decades, researchers deepened this field, synthesizing materials with micro/nano-structured surfaces driven by numerous applications other than molecular separations, such as gas storage, drug delivery, sensing and catalysis [4,5,6,7,8]. Six classes of materials are acknowledged for having a high surface area and porosity: porous carbon, porous inorganic materials, crystalline metal organic frameworks (MOFs), crystalline covalent organic frameworks (COFs) and amorphous porous organic polymers (POPs) [9]. In particular, amorphous POPs have vast potential due to to their versatile composition that provides structural diversity, including various topologies and chemical functionalities, while being lightweight and chemically stable [10]. Both COFs and POPs can be synthesized from different organic monomers, affording a high degree of chemical tunability, as well as the potential for high gravimetric and volumetric capacities due to the use of light constituent elements [11]. However, the synthetic reactions to make COFs are still limited compared with amorphous POPs, and this has hindered their broad application because they comprise regular porous surface structures with long-range order [10]. In contrast, amorphous porous polymeric materials usually have a non-uniform pore distribution in the matrix; but it is possible to easily introduce various chemical functionalities along the backbone, entailing a wide range of structures with different properties tailored on a determined application and more scalable syntheses for a better technology readiness (for commercial use) [12].

Furthermore, this synthetic flexibility/adaptability can be coupled with a high processability, to fabricate components with specific structures, designed to better exploit the extended surface area and porosity. In particular, additive manufacturing (AM) enables precise control of the macroscopic geometry of the structure, allowing the fabrication of objects not achievable with standard fabrication techniques, while the polymer characteristics define both mesostructure and morphology [13]. AM is a production technique that allows the fabrication of near-net shape three-dimensional objects, starting from a digital CAD model, in a layer-by-layer fashion, with minimal material waste and without requiring molds [14]. Polymers are the most used materials in the 3D-printing industry as a result of their diversity and ability to adapt to different printing methods, which vary for process, curing principle and the initial state of the polymer [15]. Among the numerous AM technologies, Vat Photopolymerization (VP) is the processing technique that allows better control of both architecture and microstructure, along with their mutual integration. In VP, a light, either in the form of a laser or ordinary monochromatic light rather than polychromatic, is focused on a vat containing a liquid bath comprising photosensitive species and additives, each of them playing a very specific role during printing [16]. VP 3D printing is compatible with the fabrication of a porous polymeric matrix, for instance, by block copolymer self-assembly, in which the polymeric matrix spontaneously organizes in controlled phases [17]. In this case, self-assembly can be obtained by a phase inversion process, such as non-solvent induced phase separation (NIPS), in which block copolymers act as surfactants to form emulsions, hindering the coalescence of the stabilized pores [18]. Furthermore, photopolymerizable high internal phase emulsions (HIPE) were used as inks for VP 3D printing: in this case, the photocurable resins are solidified into objects by the light irradiation, and then the aqueous phase is evaporated, leaving the porosity inside the structure [19,20]. The components of the formulation can be selected and their ratio optimized to define the porosity spatial distribution, hierarchy and geometry, together with the mechanical and functional properties [21,22].

In this context, this study aims at developing 3D printable formulations processable by a particular VP technology, Digital Light Processing (DLP), which can be used to fabricate objects with complex geometry and contemporary high porosity. DLP was selected because it combines high printing output with high precision and smooth surfaces [23]. The studied formulations are based only on commercially available chemicals, without requiring complex synthesis, making those inks scalable and suitable for potential industrialization. Morphology and chemical and mechanical properties were evaluated as function of the composition of the mixture and the printing features. The multiscale porous structure has been used then as a scaffold for the depositing of active material for CO_2_ trapping, and it has been tested as an air filter for carbon dioxide removal as proof of concept. These structures can also be envisaged as versatile platforms, easily adaptable to a wide range of applications, from air filtering to energy storage and from biological scaffolds to catalytic substrates [4,7,17,24].

## 2. Materials and Methods

### 2.1. Materials

1,6 Hexanediol (HDDA), PLURONIC F127, 2-Hydroxy-2-methylpropiophenone (photoinitiator), Pentaerythritol tetrakis(3,5-di-tert-butyl-4-hydroxyhydrocinnamate) (radical scavenger) and Sodium hydroxide were purchased from Sigma-Aldrich (Milan, Italy), Disodium 2,2′-([1,1′-biphenyl]-4,4′-diyldivinylene)bis(benzenesulphonate) (UV adsorber) was purchased from Apollo Scientific (Stockport, UK).

### 2.2. Formulation Preparation

The different formulations were prepared by mixing water and PLURONIC with a magnetic stirrer (Velp, Usmate Velate, Italy) at 1200 rpm for 1 h to obtain a correct mixing. At the end of the mixing process, HDDA, photoinitiator, Radical Scavenger and UV Absorber were added and then left for one hour sonicating to obtain a homogenous formulation. Tested formulations are summarized in Table 1.

In the second stage, Radical Scavenger and UV Absorber (0.2 phr) were added to the corresponding formulations to improve the control during the printing process. Formulations with these elements will be characterized with the abbreviation #RV, #UV or both.

### 2.3. Film Photopolymerization

The formulations were coated on glass slides as films with a thickness of 100 µm and then polymerized with a Hamamatsu Photonic (Hamamatsu, Japan) L9588-01 LC8 UV Curing Machine Spot Light Source Lightning Cure lamp with a UV light (365 nm) intensity of 25 mW/cm^2^ and an exposure time of 10 s. After the polymerization phase, the slides were immersed in water for 12 h to remove the surfactant from the polymerized formulations and the unreacted residual monomer and put in an oven to dry overnight at room temperature at a pressure of 100 mbar.

### 2.4. 3D Printing, Water Removal and Post-Processing

The printing process was performed with an Asiga (Alexandria, Australia) MAX X UV 27 DLP printer operating a 385 nm LED light source. CAD models were designed with FreeCAD open-source software (www.freecadweb.org/, accessed on 3 November 2022) and exported to .stl files to be uploaded into the proprietary printer software Asiga Composer. Various light intensities and exposure times were used, keeping the slicing layer constant at 100 μm. After the printing process, samples were immersed in an ethanol bath to remove the surfactants and the unpolymerized formulations. Subsequently, samples were subjected to UV-postcuring performed in a RobotFactory (Mirano, Italy) UV chamber equipped with a medium-pressure mercury lamp for 2 min. Finally, water extraction was performed overnight at room temperature in a vacuum oven Memmert (Schwabach, Germany) at a pressure of 100 mbar. Optimized printing parameters are reported in Appendix A.

### 2.5. 3D Scanning

The printed objects were scanned with a 3Shape (Copenhagen, Denmark) E3 scanner to verify the precision and accuracy of the print. The sample was coated with magnesium stearate in order to limit the reflection of light on the white structure, and then positioned on a platform to ensure the correct acquisition of images, subsequently digitized. The scanning was repeated 5 times to improve the quality of the acquisition. The resulting scanned image was then compared to the original CAD model by means of CA Analyzer software ( 3Shape, Copenhagen, Denmark) to generate the deviation analysis and map.

### 2.6. Morphological Characterization

Optical microscopy on UV cured thin films was performed with a Leica DFC340 FX (Leica Microsystems, Wetzlar, Germany). The images taken with the microscope were then analyzed with the image analysis software ImageJ to measure the pores size, adopting an embedded algorithm that measures the dimensions of objects in the picture by approximation to an ellipse [25]. To perform image analysis, first the full-color images collected by optical microscopy were converted into grey-scale pictures, and finally the contrast and brightness were adjusted to achieve black and white images of the microstructures. For a correct evaluation of the dimensions of the pores, the contrast and brightness parameters were maintained constant for all the pictures taken in a single experimental series. Pores analysis was performed by ImageJ (https://imagej.nih.gov/ij/, accessed on 3 November 2022), approximating the aggregates to ellipses and measuring their dimension. To improve the quality of the results, the pores located at the edges of the pictures were not considered.

Field Emission Scanning Electron Microscope (FESEM) imaging was performed with a Zeiss Supra 40 (Zeiss, Jena, Germany) on samples coated with a thin film of platinum to observe the morphology. The density variation before and after the cleaning process was measured by weighing the samples immediately after printing, and they were subsequently placed in ethanol to remove the surfactant and the unreacted monomer, and finally placed in the oven to dry for a whole night in order to remove any remaining water from the structure. The weight of the samples was then recorded again, and the density was calculated by dividing it by the volume of the sample.

### 2.7. Chemical Characterization

Photorheology was performed with an Anton Paar (Graz, Austria) MCR 302 Rheometer in plate-plate configuration, equipped with a Hamamatsu Photonics ( Hamamatsu, Japan) L9588-01 LC8 UV Curing Machine Spot Light Source Lightning Cure lamp with a UV light source with a light intensity of 25 mW/cm^2^. The gap between the lower and upper plates was kept at 150 μm where not otherwise stated, performing the tests at a temperature set to 25 °C, 1% strain and angular velocity of 10 rad/s. Ultraviolet-Visible (UV-Vis) spectroscopy was performed using a Jenway 6850 UV/Vis Single Beam Spectrophotometer (Fisher Scientific, Rodano, Italy). The spectrum region analyzed was between 200–800 nm. The Fourier Transform Infrared (FTIR) spectroscopy was performed with a Nicolet iS50 FT-IR Spectrometer (ThermoFisher Scientific, Rodano, Italy) equipped with an attenuated total reflectance (ATR) accessory. The spectra were collected on the liquid formulation and then on printed samples in the mid-IR wavenumbers range between 4.000–650 cm^−1^, with a resolution of 4.0 cm^−1^, collecting 32 scans for each spectrum. The resulting spectra were normalized using the peak of the C=O bond at 1720 cm^−1^ because it is not involved in the polymerization process. The degree of conversion was obtained by calculating the area subtended by the two spectra at the peak related to the aliphatic C=C bond at 1638 cm^−1^ before and after polymerization.

### 2.8. Mechanical Characterization

Dynamic Mechanical Thermal Analysis (DMTA) was performed by means of a Triton 2000 Triton Technology instrument (Mettler Toledo, Milan, Italy). The test was carried out at a constant frequency of 1 Hz and a displacement of 20 µm, applying an increasing temperature ramp in the range −5 °C to +150 °C with a heating rate of 3 °C/min on cuboid samples with a length of 10.00 mm, a width of 9.90 mm and a thickness of 0.94 mm.

### 2.9. Sample Decoration and Filtering Test

The dry samples were immersed in NaOH solution with different molarities (0.1 M and 1 M) and then dried at room temperature, in vacuum, for 48 h to enable precipitation onto the surface of the 3D printed object. After the treatment, the samples were tested as filters for the extraction and absorption of carbon dioxide from the air. The experimental setup for adsorption tests is based on a nondispersive IR sensor (K30 FR, manufactured by CO_2_Meter, Ormond Beach, FL, USA), which detects CO_2_ in a concentration range of 0–10,000 ppm, with a maximum sampling rate of 2 Hz and a precision of ±30 ppm ± 3% of the measured value. The electronics is integrated in the sensor and directly communicates with the control software (GASLAB, provided by CO_2_Meter) through a USB connection. A particulate filter, a hydrophobic filter and a moisture filter are used in series to avoid contamination. A miniaturized membrane pump brings the air from the sample to the sensor. The sample is placed inside a plastic cylinder, with a side open to let air to flow in. All the connections are made with PVC tubes. The scheme of the setup is shown in Appendix A.

## 3. Results and Discussion

To fabricate porous 3D printed objects, the photocurable formulation must be a water-in-oil high internal phase emulsion (HIPE), in which the hydrophobic monomers form a continuous phase after polymerization, while the pores belong to water droplets trapped in the polymeric matrix [26]. Due to the immiscibility between the monomer and water, it is crucial to obtain the internal porous structure, which in turn should possess the kinetics and mechanical properties sufficient to achieve successful 3D printing.1,6-Hexanediol diacrylate (HDDA), a bifunctional acrylate ester, has been selected as the monomer because it is a water-insoluble monomer which has limited shrinkage during polymerization, but it shows sufficient mechanical properties to avoid structural failure after water removal [27]. HDDA has a relatively fast polymerization rate, which effectively counterbalances the relatively low efficiency of the selected photoinitiator, 2-Hydroxy-2-methylpropiophenone, due to the only partial match between its absorption spectrum and the emission spectrum of the printer used (380 nm) [28]. In contrast, this radical photoinitiator was selected to avoid flocculation because it is hydrophobic and liquid; thus, it remains evenly and homogeneously distributed in the continuous phase, while other more common photoinitiators often used at these wavelengths (e.g., 2,4,6-trimethylbenzoyldiphenyl phosphine oxide, TPO) precipitate. The size and stability of the droplets in a HIPE are influenced by several factors, both process-related, such as temperature and mixing procedure, and composition-related, especially in terms of surfactant type and amount. Triblock copolymer, such as PLURONIC, has already been used as a non-ionic surfactant in the synthesis of mesoporous materials as it can interact non-covalently with the monomer without affecting polymerization and the final structure, while also enabling the formation of stable micelles that are necessary to create porosity, with a different distribution of the pore sizes according to the amount used [21].

The inks were prepared via simple mixing of the chemicals in various concentrations because the porosity of the structures is controlled by changing the water/surfactant ratio, which in turn affects the printing parameters and mechanical properties of the fabricated components. Preliminary investigations to determine the best compositions to provide the best porous microstructures were performed on UV-cured thin films. After removal of the surfactant, the films were observed by means of an optical microscope. Inks containing a quantity of water equal to or higher than the amount of HDDA after photopolymerization resulted in brittle pastes with no mechanical consistency. The large amount of water and surfactant created an inverse emulsion (oil-in-water), and the polymerization did not occur uniformly over the entire surface because monomers are arranged in small droplets surrounded by the surfactant. At the end of the photopolymerization, the tested film resulted as a viscous paste, unsuitable for the intended use. This can be observed in the microscope images of the cured polymer related to the #1.5W15.0PL and #1W10.0PL formulations (Appendix A). On the contrary, when the amount of water was lower than the number of monomers, the formulations demonstrated water-in-oil HIPE emulsions. Those, which belong to the #0.5W*** and #0.75W*** series (where *** are intended to generalize the different contents of the surfactant), after light irradiation produced a tacky-free polymeric surface, characterized by high porosity and homogeneously distributed (see examples in Figure 1a,b). The dimension and distribution of the cavities were analyzed through the imaging software ImageJ via an algorithm that approximates the shape of the pores to ellipses and measures their characteristic dimensions, such as the axis length and ellipse area (Appendix A). Despite the minor errors in right pore recognition, corrected via adjustments to the algorithm to increase the accuracy, the results consist in an average of values taken from different images for every formulation, spanning from 8 µm to 13 µm (Appendix A).

Afterwards, preliminary testing for assessing the printability of the HIPE was performed via photorheology. The polymerization kinetics of formulations containing different concentrations of water and surfactant were investigated to study the influence of the composition on HDDA crosslinking. Figure 1c reports the test for the #0.5W5.0PL formulation, which is representative of all the tested formulations that did not show differences. These tests give an approximate indication of the formulations’ behavior during the printing process, especially related to gelation time or to the extent of penetration of the radiation as a function of the light intensity. As shown in Figure 1c, the slope of the curve related to the elastic shear modulus increases immediately after turning on the lamp, indicating that the addition of water and surfactant to HDDA does not affect its polymerization kinetics. The only noticeable difference is the final average value of the storage modulus of the formulation containing water and surfactant, which is lower than the formulation containing only HDDA due to the presence of the internal phase, which contributes to a less rigid mechanical response. At last, considering that in the photorheology tests the light is shined through the quartz bottom plate, while the upper plate records the viscoelastic response, a limited light penetration and a low reactivity would cause a delay in the rising of the G’ curve that is increased by the length of the light path. For this reason, different plate distances were set and the water containing formulation was tested to understand if the presence of water and surfactant could hinder the light penetration through a defined layer thickness. As visible in Figure 1d, the penetration of the radiation was sufficient to polymerize the whole depth of a layer up to 200 µm with no delay, again indicating a good reactivity, also in the presence of water and PLURONIC.

Then DLP printing tests were performed using a holed regular cube (Figure 1a) as the reference CAD file, employing initial parameters extrapolated from the previous photorheology results. In general, DLP processing has requirements in terms of resin viscosity and part design, with the need for additional supports to anchor the object onto the build platform during printing, which are subsequently removed at the end of the print [29]. Bottom-up configuration compels the use of a transparent bottom vat to let the radiation pass, with the curing light source and projecting method being crucial for the accuracy of the printed parts [30]. In this case, all the tested HIPE formulations showed uncontrolled polymerization when processed by DLP 3D printing (Figure 2b), which resulted in clogging of the holes and unsatisfactory fabrication. Improvements can be achieved by reducing the light intensity, but this led also to incomplete and weak structures that fail during printing (Figure 2c). Therefore, it was decided to include in the photocurable formulations a water-soluble UV-absorber and a radical scavenger that allow more controlled light absorption, which may improve the resolution and obtain more complex structures [31].

The UV absorber was used to better control the penetration depth of the radiation, reducing the scattering-effect induced by the emulsion, and, thus, avoiding undesired polymerization, especially along *z*-axis [32]. In contrast, the hydrophobic radical scavenger homogeneously dispersed in HDDA prevents uncontrolled propagation of the polymerization out of the irradiated areas [33]. The following tests showed will be focused on the #0.5W5.0PL and #0.75W7.5PL formulations because those contain the higher amount of surfactant and resulted in more complex printing; nevertheless, similar behaviors were measured for the other formulations also. Photorheology tests were carried out to analyze how the addition of these two additives affects the polymerization kinetics of the formulations, and to have indications to determine the printing parameters also. For example, it can be seen in Figure 3a for the #0.5W5.0PL formulation that both the presence of the radical scavenger and the UV absorber reduce the storage modulus achieved after polymerization compared to the formulation without additives. The explanation is that even if the whole plate area is irradiated and the radicals generated from the photoinitiator are predominant, the presence of the RS diminishes their efficiency and then their ability to promote polymerization. As regards the UV absorber, it absorbs in the spectral range of emission of the printer light source, as can be seen in the UV-Visible spectra (Appendix A), and this results in a competitive absorption with the photoinitiator, inducing a decrease of photon dose available for the photoinitiator, which leads to a decrease of photoinitiation rate and, thus, slower kinetics. It can be concluded that the combined effect of the UV Absorber and the Radical Scavenger decisively influences the polymerization kinetics of the formulation, helping to confine the polymerization within the desired geometry. However, since the printer environment is less controlled than the small gap between the plates of the rheometer, the printing process will more likely necessitate a higher light intensity, as well as longer irradiation time. The results obtained from applying the new parameters to the printing process confirmed the assumptions: the object shows high accuracy, without any clogging inside hollow parts. The precision of the printing process is defined by the shape fidelity of the printed object to the reference CAD. To prove the accuracy, we performed a comparison between the digital file (Figure 3b), which is thin structures of 300 µm, and the printed object (Figure 3c) by 3D scanning. This test generates a three-dimensional model that can then be compared via software to the starting design, producing an overlay heat map that displays the deviation (Figure 3d). It results in an average variation from the original piece of around 50 µm, indicating high fabrication precision, visually reported as the color green in Figure 3d. To further evaluate the printing precision, a layer-by-layer comparison between the planar section of the CAD reference and the scanned reconstruction allows evaluation of the deviation, which produces very limited results (Appendix A). Irregularities and divergences can be attributed to several factors, such as the printer resolution, printing errors, scanner precision and difficulty properly detecting internal areas of the structure, in addition to the uneven distribution of magnesium stearate powder on the surface of the object, which is used to improve scanning efficiency, as well as the presence of a support to enable the correct acquisition of the images (Appendix A).

After a macroscopic evaluation of the quality of the architecture, a morphological characterization of the microstructure was performed by means of Field emission scanning electron microscope (FESEM) to observe the details of the mesoporosity in the printed structures and to see how the amount of surfactant affects the internal structure of the sample. It is evident that a greater concentration of surfactant induces structures with a higher porosity, which appears more uniform along the entire structure (Figure 4a,b). Larger pores have spherical geometry with a sponge-like surface, derived from the vesicles of monomers formed during the emulsion process (Appendix A). However, the cavities appear to be arranged in a hierarchical arrangement, in which the pores develop from the surface towards the inside, only in objects fabricated with the formulation containing the highest concentration of surfactant. A more distributed and interconnected porosity within the structure is achieved due to the internal channels created by the contact among the droplets of the internal phase of the precursor ink (Figure 4c,d). Interconnected porosity (Appendix A) allows an efficient surfactant extraction, which can be indirectly evaluated via density variation between cubes printed with only HDDA compared to objects with the same regular geometry printed using the formulation under exam before and after the cleaning process. The quantitative analysis was performed by measuring the weight loss after an ethanol bath to remove the surfactant and unreacted monomer, followed by an overnight drying. The average density decreased from an initial 1.027 g/cm^3^ to 0.963 g/cm^3^, resulting in 6% weight loss with a Standard Deviation (σ) equal to 0.0356, confirming the presence of a partially interconnected internal hierarchical porosity.

The presence of the pores obviously affects the mechanical strength of the printed objects, which also depends on the acrylate conversion. Fourier Transform Infrared (FTIR) Spectroscopy was used to investigate the degree of conversion of the C=C double bonds relative to the ratio of the areas subtended by the peaks located at 1638 cm^−1^ in the absorption spectrum, relative to the wavenumber for the aliphatic double bond before and after curing (Figure 4e) [34]. The results show that as the amount of surfactant inside the formulation increases, the degree of conversion slightly decreases if compared to a formulation containing only HDDA and photoinitiator (Appendix A). This can be attributed to the interference brought by the droplets surrounded by an amphiphilic polymer, partially compatible with the matrix, which interposes HDDA chains, limiting the effective interaction and crosslinking and leaving a larger quantity of unpolymerized monomer and unreacted functional groups that are removed during washing.

The partially detrimental effect of the presence of droplets and surfactant in the matrix, proven by the decrease of both the storage shear modulus and the degree of conversion, as shown in the photorheology and FTIR spectroscopy, respectively, was confirmed by the Dynamic Mechanical Thermal Analysis (DMTA). The tests were performed at a constant strain and increasing temperature at a constant heating rate on parallelepipeds printed with the complete formulation proposed and compared to samples made only of HDDA as reference. The presence of the internal phase leads to a decrease in the crosslinking density, as evidenced by the decrease of the modulus values with the increase of the concentration of additives inside (Figure 4f and Appendix A). It can be noted that there is a shift in the peak of the tan δ in Figure 4f, related to a slight increase in the glass transition temperature, as well as an increase in the FWHM, which can be attributed to a larger crosslinking inhomogeneity.

At last, the 3D-printed polyHIPE were tested as scaffold for carbon dioxide capture in an air filter, designing a structure composed of staggered strips to enable correct filtering of the air flow (Appendix A). Filters were printed with the #0.75W7.5PL#RS#UV formulation, characterized by the best internal morphology for the scope, especially when coupled with an appropriate architecture for the purpose (Figure 5a). The printed structures were treated with Sodium Hydroxide by immersing them in two different solutions with different molarities for 48 h in vacuum to let the soda precipitate onto the surface while removing the solvent in order to study how the different NaOH concentrations affect the absorption (comparison with pristine surface in Appendix A). The NaOH deposits derived from the 0.1M solution arrange into leaflets, which will act as a trap for the carbon dioxide particles present in the air (Figure 5b), while the treatment in the 1M solution results in the conformal (Figure 5c) formation of agglomerates covering a larger portion of the surface due to the higher soda concentration in the solution (Figure 5d). The structures were then placed in a testing chamber, showing absorption of carbon dioxide present in the air. When connected, the CO_2_ values immediately drop from the starting value of ~350 ppm (environmental concentration of CO_2_) to values around 250 ppm due to soda absorption. After a certain time, both the filters saturate (i.e., the plateau value in Figure 5e after 2500 s), and the measured CO_2_ values come back to the initial value. This indicates that all the NaOH nanoparticles that decorate the surface of the filters have reacted with the gas. Filters treated in 0.1M solution absorbs for a very long time (>2500 s, Figure 5e), while the ones treated at 1M saturate much earlier (around 1000 s, Figure 5f). Consequently, this second treatment leads to a total absorption 15 times lower than in the specimen treated with solution at 0.1 M, even if at a faster rate. This seems to suggest that more uniform deposition coverage of the surface is not sufficient to provide efficient absorption, as a more compact and dense salt deposit negatively affects the efficiency, saturating in a shorter time. In this direction, although feasibility was demonstrated here, more tests are necessary to optimize the deposition parameters, which will be studied in future investigations.

## 4. Conclusions

Successful 3D printing of geometrically complex and porous structures was achieved with high shape fidelity to the original geometry, exploiting the characteristics of the polyHIPEs. The ink for a precise printing was based on a readymade formulation composed of commercially available materials without requiring complex synthesis, which is easily scalable and suitable for potential industrialization. Here, various formulations were tested, indicating that only water-in-oil emulsions were 3D printable, i.e., when the amount of water is lower than the number of photocurable monomers. However, printable formulations showed only slight differences, except for the pores dimension.

Our solution combines the enormous potential of an all-in-one fabrication technique—such as the DLP 3D-printing method with a polymerization-induced porosity creation process, using a surfactant as the templating agent—to produce objects with accurate complex structure characterized by an enhanced surface-to-area-to-volume ratio and interconnected porosity. Light-activated AM enables control of the geometry at various scales because the formulation composition determines the microstructure and the surfactant/solvent combination, and ratio defines the pores morphology while the CAD design and printing parameters determine the macroscopic architecture. The optimization of these porous scaffolds must go through precise control of the interactions between morphology, materials and fabrication technology across different scales. In particular, the presence of the surfactant, as well as the highly concentrated internal inert phase, can have a negative impact on both polymerization kinetics during printing and mechanical properties of the final object. Additionally, scattering in the emulsion can hinder printing, but this can be solved adding water-soluble dyes that cooperate with radical scavengers to prevent polymerization outside the designated areas. The polyHIPE proved to be a suitable solution as scaffold for the deposition of active material for CO_2_ trapping, enabling efficient air filtering. The proposed polymeric scaffold is a versatile platform that can be shaped into an ad-hoc design tailored to the specific application, gathering complexity from the macro- to the microscale. These findings can open interesting perspectives in numerous applications, such as catalytic chemistry, electrochemical energy devices, sensors and filters.

## Figures and Tables

**Figure 1 polymers-14-05265-f001:**
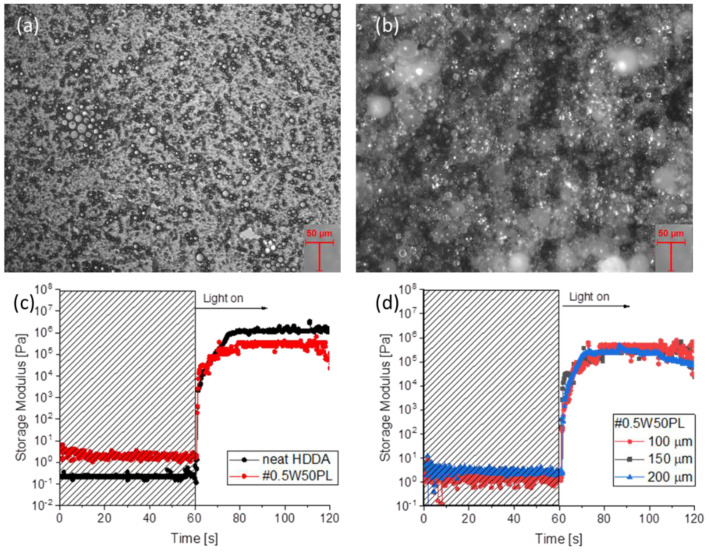
Optical microscopy images (20×) of #0.5W5.0PL (**a**) and #0.75W5.625PL (**b**) thin films. Photoreology measurements comparing neat HDDA and #0.5W5.0PL formulations (**c**) Photoreology measurements of #0.5W5.0PL at different plates distance (**d**).

**Figure 2 polymers-14-05265-f002:**
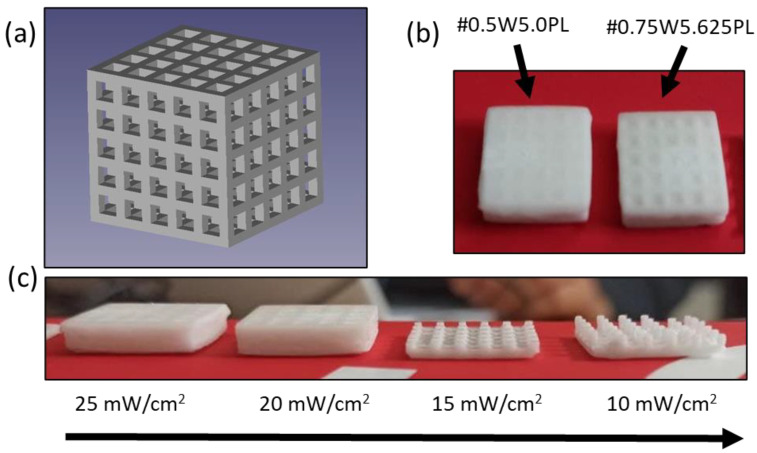
CAD file used for testing printability of the formulations, the cube is 2 × 2 × 2 cm^3^ (**a**). Preliminary printing experiments of #0.5W5.0PL (left) and #0.75W5.625PL (right) formulations (**b**). Optimization testing reducing light intensity from 25 mW/cm^2^ (left) to 10 mW/cm^2^ (right) in #0.75W5.625PL formulation (**c**).

**Figure 3 polymers-14-05265-f003:**
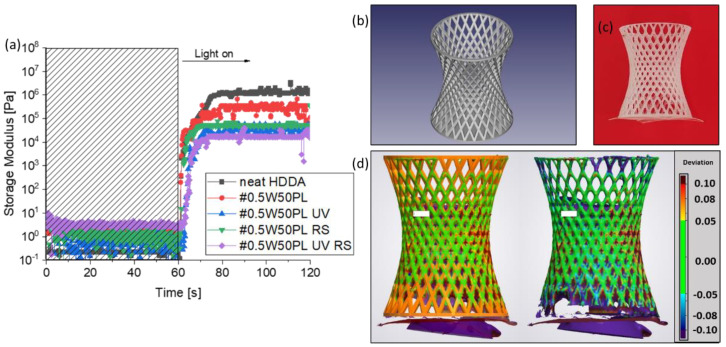
Photoreology tests (**a**) CAD of the complex printed structure (base radius 1 cm, height 4 cm) (**b**) Picture of the correspondent 3D printed part (**c**) and 3D scanner comparison (**d**).

**Figure 4 polymers-14-05265-f004:**
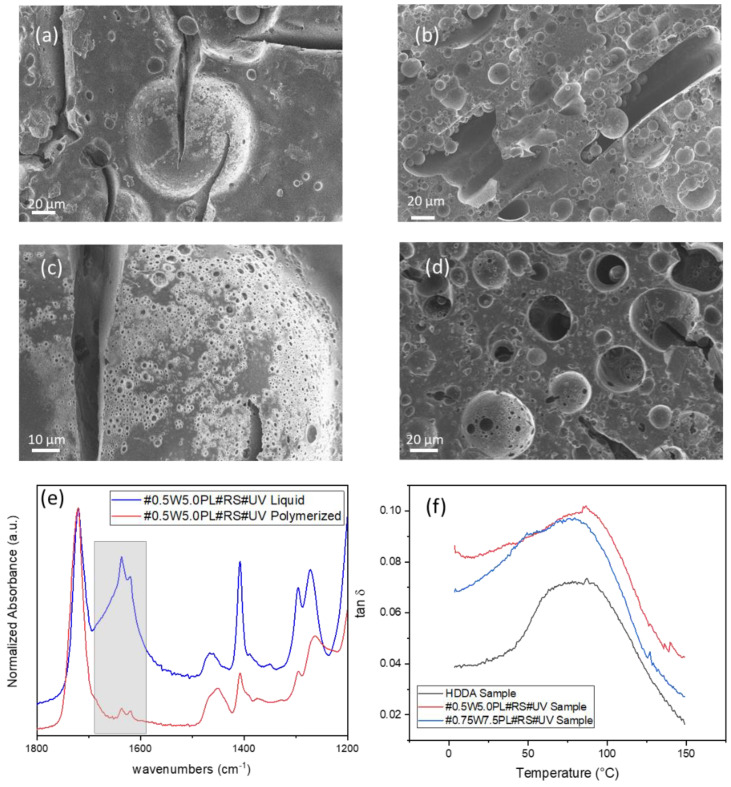
SEM images #0.5W5.0PL#RS#UV (**a**), #0.75W7.5PL#RS#UV (**b**) with a magnification M = 700x. SEM magnification of #0.5W5.0PL#RS#UV (**c**), and #0.75W7.5PL#RS#UV (**d**). Example of FT-IR on liquid formulations and printed structure. The grey box highlights the C=C peak range used to calculate conversion (**e**) DMA curves of different materials (**f**).

**Figure 5 polymers-14-05265-f005:**
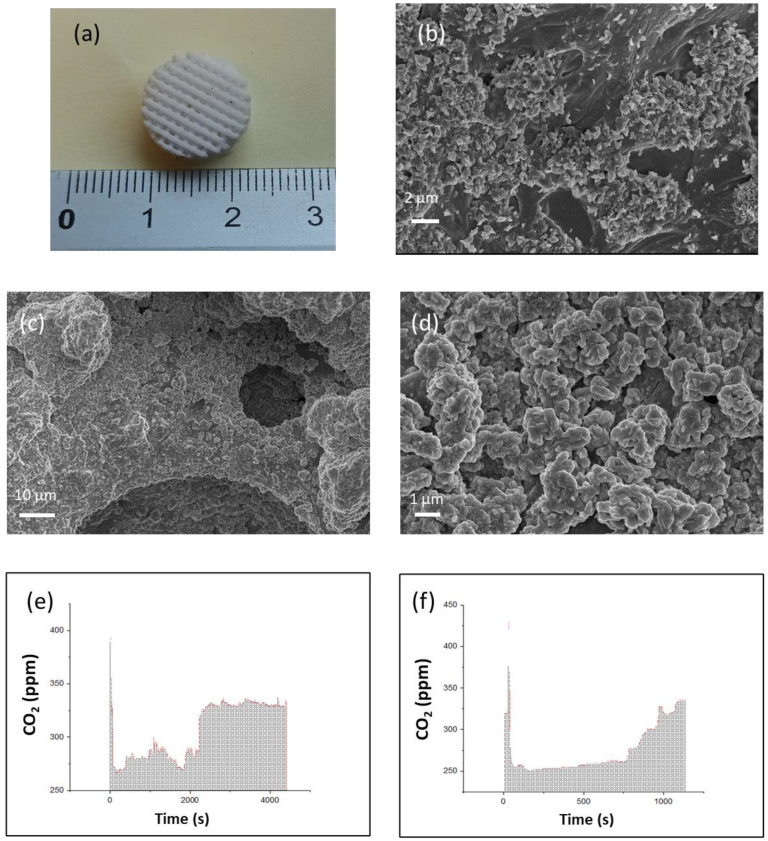
Filter-like structure 3D printed with #0.75W7.5PL#RS#UV formulation (**a**) SEM images 3D printed parts after precipitation of NaOH nanoparticles at 0.1 M (**b**) and 1 M (**c**). Magnification of deposited NaOH nanoparticles (**d**). Test of CO_2_ absorption on filter-like structures after precipitation of Nps from 0.1 M (**e**) and 1 M (**f**).

**Table 1 polymers-14-05265-t001:** List of the tested photocurable formulations.

HDDA (phr)	Photoinitiator (phr)	Water (phr)	PLURONIC (phr)	#NAME
100	4	50	2.5	#0.5W2.5PL
100	4	75	3.75	#0.75W3.75PL
100	4	100	5	#1W5.0PL
100	4	150	7.5	#1.5W7.5PL
100	4	50	3.75	#0.5W3.75PL
100	4	75	5.625	#0.75W5.625PL
100	4	100	7.5	#1W7.5PL
100	4	150	11.25	#1.5W11.25PL
100	4	50	5	#0.5W5.0PL
100	4	75	7.5	#0.75W7.5PL
100	4	100	10	#1W10.0PL
100	4	150	15	#1.5W15.0PL

## Data Availability

Data are available from the authors upon request.

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
