# Peer review of "Photocurable 3D-Printable Systems with Controlled Porosity towards CO2 Air Filtering Applications"

_polymers, 2022, doi:10.3390/polym14235265_

Round 1

Reviewer 1 Report

The paper deals with the problem is CO2 trapping structures made by 3D printing, what is very interesting and up-to-date. In my opinion the work is well presented and needs only a small improvement:

 - please add in abstract the quantitative results of the study,

 - please specify if the 3D printing resolution was considered as well as the change in curing and postprocessing parameters and can they influence the results,

  - how many samples were tested, did authors observe any distribution or randomization of results - please add a comment about it,

- the paper should be edited dou to journal's guidelines.

Reviewer 2 Report

The work presents a photocurable ink based on 1,6 Hexanediol diacrylate containing a high internal phase emulsion (HIPE), employing PLURONIC F-127 as a surfactant. Different parameters were suggested to study the effects on the morphology of the pores, the printability, and the mechanical properties. However, the novelty has not been emphasized yet. The logic of the presented results should be improved for better understanding. Therefore, the manuscript needs major revisions to be qualified to publish in Polymers. Below are the specific points that require attention.

1)     The limitations of current studies should be described to emphasize the authors’ research contributions and novelty.

2)     The name #0.75W56.25PL in table 1 seems to be wrong. In addition, please use the same format names in the whole manuscript because some sudden names such as #0.5W_25PL, #0.75W_375PL, #0.5W50PL, and #0.75W_75PL make the reader hard to follow.

3)     What do the term “(#0.5W***, #0.75W***)” in “On the contrary, HIPE formulations based on a water-in-oil emulsion (#0.5W***, #0.75W***) resulted in a continuous solid surface after light irradiation, characterized by high porosity homogeneously distributed in the film (Figures 1a and 1b)” mean?

4)     What is the algorithm used in ImageJ to analyze the dimension and distribution? Please explain in more detail the principle.

5)     The authors provided 12 tested photocurable formulations in their study, but the presented results are hard to follow, with few showed tested formulations. For example, the pores' average diameters of only four formulations (#0.5W_25PL, #0.75W_375PL, #0.5W50PL, and #0.75W_75PL) were presented in Table S2. On the other hand, the authors showed results of the names #0.5W5.0PL and #0.75W5.625PL in Figures 1 and 2, while they showed the results of the names #0.5W50PL and #0.75W75PL in Figures 4. Please re-organize the results for better understanding.

6)     What is the depth of the penetration of the radiation?

7)     Please discuss the effect of different plates distance on the photorheology measurements.

8)     The statement “Thus, to improve the resolution additives that can control light absorption and polymerization reaction were added” should be improved for better understanding.

9)     Please indicate the large bubbles in Figure S2.

10) The legend in figure S4 is hard to see. Which formulation was used in Figure S4? Please also indicate the peak to emphasize the differences.

11) The texts in figure S5 are hard to see.

12) Please discuss in more detail the obtained results in Figure 3d.

13) After the description of figure 3, the authors described figures 5a and 5b before the description of Figure 4: “It is evident that a greater concentration of surfactant induces structures with a higher porosity, which appears more uniform along the entire structure (Figures 5a and 5b). This makes the reader hard to follow. Please re-organize the content of the manuscript.

14) Please indicate the pores’ interconnection in Figure S8.

15) Please indicate the peaks in figure 4a to show the differences.

16) Please add scale bars for Figures 2b, 2c, 3c, S2, S3, S7, S8, and S11.

17) Please add the captions for Figures 1d, 4e and 4f.

18) Please discuss more Figures 5e and 5f.

19) The discussion part seems incomplete: “Authors should discuss the results and how they can be interpreted from the perspective of previous studies and of the working hypotheses. The findings and their implications should be discussed in the broadest context possible. Future research directions may also be highlighted”. Please improve this part.

20) English writing should be improved for better understanding.

21) Many grammatical mistakes, such as “CO2”, “cm2”, “3. D”, “cm-1”, and spaces between values and units, should be corrected.

Round 2

Reviewer 2 Report

The quality of the work was significantly improved. There are two minor things to improve the manuscript. The manuscript, after minor revisions, is qualified to publish in Polymers. Below are the specific points:

1)     The color and the size of the arrows in Figure S8 should be revised to make it clearer.

2)     A space between removal and [27] should be added.